# Genetic Ablation and Guanylyl Cyclase/Natriuretic Peptide Receptor-A: Impact on the Pathophysiology of Cardiovascular Dysfunction

**DOI:** 10.3390/ijms20163946

**Published:** 2019-08-14

**Authors:** Kailash N. Pandey

**Affiliations:** Department of Physiology, Tulane University Health Sciences Center, School of Medicine, New Orleans, LA 70112, USA; kpandey@tulane.edu; Tel.: +1-(504)-988-1628; Fax: +1-(504)-988-2675

**Keywords:** atrial natriuretic peptide, guanylyl cyclase/natriuretic peptide receptor-A, gene-knockout, gene-duplication, hypertension, congestive heart failure

## Abstract

Mice bearing targeted gene mutations that affect the functions of natriuretic peptides (NPs) and natriuretic peptide receptors (NPRs) have contributed important information on the pathogenesis of hypertension, kidney disease, and cardiovascular dysfunction. Studies of mice having both complete gene disruption and tissue-specific gene ablation have contributed to our understanding of hypertension and cardiovascular disorders. These phenomena are consistent with an oligogenic inheritance in which interactions among a few alleles may account for genetic susceptibility to hypertension, renal insufficiency, and congestive heart failure. In addition to gene knockouts conferring increased risks of hypertension, kidney disorders, and cardiovascular dysfunction, studies of gene duplications have identified mutations that protect against high blood pressure and cardiovascular events, thus generating the notion that certain alleles can confer resistance to hypertension and heart disease. This review focuses on the intriguing phenotypes of *Npr1* gene disruption and gene duplication in mice, with emphasis on hypertension and cardiovascular events using mouse models carrying *Npr1* gene knockout and/or gene duplication. It also describes how *Npr1* gene targeting in mice has contributed to our knowledge of the roles of NPs and NPRs in dose-dependently regulating hypertension and cardiovascular events.

## 1. Introduction

Almost 40 years ago, de Bold and coworkers established that the extracts of atria contain natriuretic, diuretic, and vasorelaxant activity [1]. This pioneering discovery led them to identify and characterize atrial natriuretic peptide (ANP) in the heart [1,2]. Natriuretic peptides (NPs) exhibit critical functions in the control of renal, cardiovascular, endocrine, neural, and skeletal homeostasis [2,3,4,5,6,7,8,9]. The natriuretic, diuretic, and vasorelaxant effects of ANP, the first member of the NP hormone family to be discovered, are largely directed at lowering blood pressure (BP) and cardiovascular homeostasis [5,6,10]. Subsequently, two other NPs, brain natriuretic peptide (BNP) and C-type natriuretic peptide (CNP), were discovered. Each of these peptide hormones is encoded by a separate gene [6,11,12,13]. The three natriuretic peptides (ANP, BNP, and CNP) have highly homologous structure, but distinct sites of synthesis and secretion. Both ANP and BNP are predominantly synthesized in the heart. ANP concentrations range from 50- to 100-fold higher than BNP [14]. CNP, which is largely synthesized in endothelial and neuronal cells is not much released into the circulation [15]. All three natriuretic peptide hormones have highly homologous structure, bind to specific cell-surface cognate receptors, and exert distinct biological functions [4,6,16]. 

Endogenous peptide hormones, including ANP, BNP, CNP, and urodilatin (Uro), are considered to have integral roles in hypertension and cardiovascular regulation [2,4,6,7,17]. ANP and BNP not only regulate BP, but also maintain antagonistic action in response to the renin–angiotensin–aldosterone system (RAAS) [4,6,17,18,19]. They also have effects on endothelial cell function, cartilage growth, immunity, and mitochondrial biogenesis [7,10,20,21,22,23]. ANP and BNP are used in hospitals to diagnose the etiologies of shortness of breath, cardiovascular dysfunction, and congestive heart failure in patients with emergency conditions [24]. The discovery of structurally related natriuretic peptides indicated that the physiological control of BP and cardiovascular homeostasis is complex and multifactorial. 

Three subtypes of NP receptors have been identified and characterized: natriuretic peptide receptor-A (NPRA), receptor-B (NPRB), and receptor-C (NPRC). NPRA and NPRB both contain intrinsic cytosolic guanylyl cyclase (GC) catalytic domains, produce intracellular second messenger cGMP. These receptors are also referred to, respectively, as guanylyl cyclase-A (GC-A) and guanylyl cyclase-B (GC-B) [3,6,16,25,26]. Thus, both NPRA and NPRB have also been respectively designated as guanylyl cyclase/natriuretic peptide receptor-A (GC-A/NPRA) and guanylyl cyclase/natriuretic peptide receptor-B (GC-B/NPRB). GC-A/NPRA is a principal locus involved in the regulatory action of ANP and BNP [6,10,27,28]. 

Biochemical, immunohistochemical, and molecular studies have suggested that ANP, BNP, CNP and their three distinct receptor subtypes (NPRA, NPRB, and NPRC) have widespread tissue and cellular distributions, indicating the occurrence of pleotropic actions at both systemic and local levels (Figure 1). Both ANP and BNP activate NPRA, which produces second-messenger cGMP in response to hormone binding. CNP activates NPRB, which also generates cGMP, but all three NPs indiscriminately bind to NPRC, which lacks a GC domain and does not produce cGMP [29,30,31]. The ligand–receptor complexes of NPRA, NPRB, and NPRC are rapidly internalized in the intracellular compartments and a majority of ligand-bound receptors are degraded in the lysosomes; however, a small population of receptors recycle back to the plasma membrane [6,32,33,34,35,36].The combined cellular, biochemical, molecular, and pharmacological aspects of NPs and their prototype cognate receptors have demonstrated hallmark physiological and pathophysiological effects, including renal, cardiovascular, neuronal, and immunological effects that are important in health and disease [6,9,10,23,37,38,39]. 

Gaining insight into the intricacies of the ANP-BNP/NPRA/cGMP signaling system is of pivotal importance for understanding both receptor biology and the pathology of disease conditions arising from abnormal hormone–receptor interplay. It has been demonstrated that binding of ANP and BNP to the extracellular ligand-binding domain of NPRA causes a conformational change, thereby transmitting the signal to the intracellular carboxyl-terminal GC catalytic region, which produces second-messenger cGMP from substrate GTP in target cells and tissues [16,25,30]. Earlier studies focused on elucidating the biochemical and molecular nature of the function of GC-A/NPRA [10,38,39,40]. Both cultured cells in vitro and gene-targeted (gene-knockout and gene-duplication) mouse models in vivo have been used to gain understanding of the NPs and their receptors to delineate their normal and abnormal control of pathophysiological processes in hypertension and cardiovascular disease states. Although there has been a strong focus on the functional roles of ANP, BNP, and NPRA in cardiovascular, endocrine, and renal homeostasis, in-depth studies are still needed to identify their potential molecular targets in cardiovascular diseases states. It is expected that ongoing and future studies on natriuretic peptides and their receptors will provide with new therapeutic targets and novel loci for the control and treatment of hypertension, cardiovascular events, and neurological dysfunction.

## 2. Historical Perspectives and Background

An increase in atrial stretch triggers the release of atrial natriuretic peptide (ANP) and brain natriuretic peptide (BNP), which largely control homeostasis of extracellular fluid, blood volume, and BP, and cardiovascular homeostasis under both normal and pathophysiological conditions [1,2,4,6,7,17,41,42,43,44]. BNP exhibits a pattern of action similar to that of ANP, but there are differences between ANP and BNP with regard to pro-hormone processing, storage, and release [17,45,46,47,48,49]. Although circulating levels of both ANP and BNP are elevated in patients with cardiovascular and renal disease, the infusion of BNP in patients with congestive heart failure has shown more beneficial effects [50,51]. NPRA is considered to be a primary ANP- and BNP-signaling molecule because major cellular and physiological responsiveness of these peptide hormones i**s** mimicked by cGMP analogs [4,6,14,43,52]. 

Efforts to design chimeric natriuretic peptides has led to the synthesis of biologically active peptide molecules, which represent single-chemical entities that combine the structural and functional properties of two different natriuretic peptides [53]. These chimeric natriuretic peptides, which exhibit the actions of two different natriuretic peptides, often reduce adverse biological effects. One such chimeric peptide molecule, which is derived from CNP and *Dendroaspis* natriuretic peptide (DNP) is known as CD-NP; it is produced by the fusion of 22-amino acid residues of CNP with the 15-amino-acid residues of C-terminus of DNP [53]. The chimeric hormone CD-NP exhibits vasorelaxant properties, effectively reduces cardiac volume overload, and exerts renal and cardiovascular protective effects. At the same time, the 15-amino-acid C-terminus of DNP is extremely resistant to neutral endopeptidase (NEP), making CD-NP a more stable and desirable peptide molecule than are naturally occurring natriuretic peptides hormones [53,54].

Experimental evidence suggests that the ANP-BNP/NPRA system has an important role in renal function, which is directed toward lowering BP to reduce the heart load [9,10,17,38,55]. Determining the physiological role of ANP by studies using large infusions of hormone has been difficult because of the interdependent nature of many organ-system responses and the counter-regulatory effect that is often evoked. Earlier, several approaches were used in attempts to define the physiological responses of ANP. For example, cardiac appendectomy was used to prevent ANP release. However, the missing normal cardiac function resulted in a lack of the physiological reflexes normally induced by atria [56]. Another approach used the monoclonal antibody against circulating ANP, but problems occurred because of the nonspecific effects of the antigen–antibody complexes. One classical approach was to use receptor antagonists specifically to inhibit the signaling pathway of NPRA and block cGMP production. Although two compounds, A71915 and HS-142-1, were shown to diminish the effect of ANP, neither completely inhibited NPRA [6,17,57,58,59,60]. 

The advent of molecular biology opened the way for genetic analysis, which is a faster and more accurate way to evaluate physiological and pathophysiological functions. Genetic studies of live mice have been accomplished by gene targeting (gene-knockout and gene-duplication), in which the desired genetic changes are specially made in mouse germ lines. This allows examination of the effect of genes of interest, such as those controlling hypertension and cardiovascular dysfunction [61]. Gene titration (gene-targeting) in mice is a novel method for use in investigations that have either ablated (gene-knockout/gene-disruption) or increased the number of gene copies (gene-duplication) at the normal chromosome location [62]. Gene targeting, including gene-ablation and gene-duplication, can lead to the production of offspring having zero-copy (0/0; −/−), one-copy (1/0; +/−), two-copies (1/1; +/+), three-copies (2/1; ++/+), or four-copies (2/2; ++/++) of the target gene of interest. Genetically modified animals have provided excellent models for the study of gene-dose-dependent physiological responses in vivo [63]. 

Studies with *Npr1* (encoding NPRA) gene-knockout mice have demonstrated that a deficiency in NPRA increases BP by 35–45 mmHg and causes renal, vascular, and cardiac dysfunction that leads to hypertensive heart disease in mice, much like that observed in untreated hypertensive patients [64,65,66,67,68,69]. On the other hand, increased expression (gene-duplication) of NPRA in mice significantly reduces BP and increases second-messenger cGMP corresponding to the level of *Npr1* gene copy number in both male and female mice [70,71,72,73,74,75,76,77]. 

## 3. Ablation of *Nppa* and *Nppb* Triggers Hypertension and Cardiovascular Dysfunction

Gene-targeting strategies in mice have provided novel approaches to the study of physiological responses corresponding to gene dosages in vivo [10,38,78,79]. Experimental evidence has led to the notion that the ANP and BNP system is important in regulating BP by its direct vasodilatory effect, as well as by natriuretic and diuretic responses and the antagonistic actions of the RAAS [4,6,9,17,52]. Studies using gene disruption of *Nppa* (coding pro-ANP) and *Nppb* (coding for pro-BNP) have shown that ANP- and BNP-deficient genetic strains of mice exhibit a defect in ANP and BNP synthesis that can cause hypertension and cardiovascular dysfunction in homozygous null mutant mice with no circulating or cardiac ANP and BNP, respectively [80,81]. Therefore, genetic defects that reduce the activity of the NPs system can be considered important candidates as contributors to essential hypertension and cardiovascular dysfunction. Previous findings showing elevated BP in genetic mouse models with loss of ANP or BNP expression and circulating hormones have provided strong support for a physiological and pathophysiological role of the NPs hormone system in regulating BP and cardiovascular homeostasis [80,81,82]. 

### 3.1. Gene Ablation of Nppa Increases Blood Pressure

Gene-targeting studies of *Nppa* and *Nppb* in mice carrying gene-knockout have presented strong evidence of the physiological roles of ANP and BNP hormones and their signaling system in hypertension and cardiovascular dysfunction [80,81,82]. In *Nppa* homozygous null mutant (*Nppa*^−/−^) mice on standard- or intermediate-salt diets, BP was elevated by 8–12 mmHg. Heterozygous (*Nppa^+/−^)* mice on a standard-salt diet showed a normal amount of circulating ANP and normal BP. On the other hand, heterozygous *Nppa*^+/−^ mice on a high-salt diet became hypertensive; their BP was elevated by 20–28 mmHg [80]. These studies demonstrated that a mutation in the *Nppa* allele could lead to salt-sensitive hypertension even though the plasma ANP level was not significantly reduced. Thus, genetically reduced production of ANP could cause salt-sensitive hypertension. Transgenic mice overexpressing ANP developed sustained hypotension. Also, their mean arterial pressure (MAP) was 25–30 mmHg lower than that in nontransgenic control animals [82,83]. Overexpression of ANP in hypertensive mice significantly lowered systolic BP; similarly the somatic delivery of *Nppa* in SHR exhibited a sustained reduction in BP, suggesting that ANP gene therapy could be used to treat hypertensive humans [84,85]. 

A significant positive correlation between plasma ANP levels and pulmonary vascular resistance in normal individuals and patients with heart and lung disease has also been found [86,87]. Moreover, it has been demonstrated that after chronic hypoxic exposure transgenic mice overexpressing ANP, as compared to nontransgenic control mice, develop very low right ventricular hypertension and vascular remodeling [88]. As compared to control animals, mice lacking functional *Nppa* (*Nppa^−/−^*) developed more severe pulmonary hypertension and right ventricular hypertrophy with chronic hypoxia [89,90,91,92]. These early findings supported the notion that *Nppa* has a critical role in modulating hypertension and right ventricular hypertrophy in hypoxic conditions. 

ANP transgene in mice has been shown to be capable of maintaining sodium balance in mice on a very low-salt diet without evidence of salt depletion, suggesting an independent action of ANP on BP and renal hemodynamic functions [82,93]. A similar conclusion was advanced based on findings in *Nppa* gene-knockout mice, in which relative hypertension was observed after mice had been on a high-salt diet for one week [80]. Further studies indicated that the lack of ANP action in *Nppa* gene-knockout mice might be due to the inability of ANP to properly regulate plasma renin and angiotensin II (Ang II) levels, which result in the development of an additional component of salt-sensitive hypertension [94]. A deficiency in endogenous ANP might play a regulatory role in the hypertensive disease state; however, the genetic basis of salt-sensitive variants of hypertension still remain controversial. It is possible that sensitization of arterial BP to dietary salt may develop as a consequence of defective functional alterations in both salt- and BP-regulating mechanisms. As a result, the genetic deficiency in ANP synthesis might play a critical role in the regulation of BP homeostasis. Monoclonal antibodies against ANP were often used to block the endogenous action of ANP in spontaneously hypertensive rats (SHR), which resulted in accelerated development of hypertension in these animals compared with Wistar Kyoto (WKY) rats [95]. 

### 3.2. Role of Nppa/Nppb in Cardiac Remodeling and Dysfunction

ANP and BNP exert cardioprotective effects not only as circulating endocrine hormones, but also as local autocrine and paracrine factors. Plasma levels of both ANP and BNP are markedly elevated under the pathophysiological conditions of cardiac dysfunction, fibrosis, hypertrophy, pulmonary embolism, and congestive heart failure [66,67,96,97,98,99,100,101]. The expression of both ANP and BNP is greatly increased in proportion to the severity of the pathophysiology of cardiac remolding and disorders in the experimental animal models and humans [7,97,100,102,103]. Levels of ANP and BNP are greatly increased in the cardiac tissues and plasma of patients with hypertensive heart disease and congestive heart failure. In humans, high plasma levels of ANP and BNP tend to predict congestive heart failure and both of these peptide hormones appear to reduce the pre- and after-load of the heart [96,97,102,104]. In patients with severe congestive heart failure, both ANP and BNP levels increase, but BNP levels rises 25-fold to 50-fold higher than do ANP levels [9,105]. Both *Nppa* and *Nppb* genes are overexpressed in hypertrophied hearts, suggesting that both the autocrine and/or paracrine effects of ANP and BNP predominate and endogenously protect against the maladaptive factors of pathological cardiac hypertrophy and dysfunction [10,38,66,67,73,77,96,100,106,107]. 

Ventricular expression of *Nppa* and *Nppb* is more closely associated with local cardiac hypertrophy and fibrosis [66,67]. Since the half-life of plasma BNP is longer than that of ANP, diagnostic evaluations of NPs have predicted and favored BNP as a critical indicator of cardiovascular dysfunction and congestive heart failure in emergency-room patients with chest pain [97]. However, NT-proBNP also seems to be a stronger indicator of the risk of congestive heart failure and cardiovascular disorders [108,109,110,111]. The expression levels of *Nppa* and *Nppb* are greatly stimulated in hypertrophied and failing hearts [66,106,112]. Interestingly, mutation in the *Nppa* promoter has been shown to trigger cardiac hypertrophy in WKY and Wistar Kyoto-derived hypertensive (WKYH) rats [113]. 

Earlier studies suggested that expression of *Nppa* is one of the critical responders to different hypertrophic stimuli in the heart [114]. Thus, the induction of *Nppa* gene expression can be considered to be a pathognomonic indicator of cardiac hypertrophy that seems to be conserved across species [115]. Similarly, *Nppb* gene expression and BNP levels are most robustly increased in patients with cardiac hypertrophy [116,117]. Increased cardiac mass and hypertrophic growth are characterized by increased synthesis of contractile and matrix proteins and reactivation of embryonic genes [115,118]. It is thought that the function of ANP and BNP is to reduce load-specific changes in the heart by increasing vasodilatory responses and enhancing natriuretic and diuretic effects, leading to decreased body fluid and blood volume [4,7,9,17]. It has been found that ANP transgene overexpression in mice results in decreased heart weight and prevents hypertrophic responses [88]. Mice lacking *Nppb* did not develop cardiac hypertrophy or increased BP, but did show an anti-fibrotic role for BNP [81]. It is thought that the direct effect of ANP and BNP can be mimicked by the use of agents such as 8-Br-cGMP or zaprinast, which increase intracellular second-messenger cGMP levels and have anti-hypertrophic and apoptotic effects in the heart [38,55,119,120,121,122]. 

## 4. Genetic Disruption of *Npr1* and Pathophysiology of Hypertension and Cardiovascular Events

Much of the early work on NPRA was directed at determining the biochemical character and cellular signaling of the receptor molecule; later studies were aimed at delineating the structure–function relationship and physiological and pathophysiological significance of NPRA in regulating hypertension and cardiovascular events. Although the early studies provided a wealth of information about the biochemical and molecular properties of NPRA, the mechanisms regulating the physiological functions of the ANP-BNP/NPRA system in vivo remained less understood. Development and analyses of gene-targeted (gene-knockout and gene-duplication) mutant mouse models of *Npr1* allowed a more direct assessment of the physiological function of *Npr1* in intact animals in vivo (Table 1). Disruption of *Npr1* encoding NPRA increases BP and causes hypertensive heart disease in null mutant mice; this heart disease is similar to that seen in patients with untreated hypertensive heart disease [6,9,71,123]. In contrast, increased expression of *Npr1* in gene-duplicated mice significantly reduces BP and cardiovascular events and increases GC activity and intracellular second-messenger cGMP, which corresponds to *Npr1* gene-copy numbers in vivo [64,71,72,73,74,77]. Thus, complete absence of NPRA protein in mice causes hypertension and leads to cardiac hypertrophy; also, particularly in males, the absence of NPRA protein leads to lethal vascular complications and aortic dissection [64,66,100,106]. The resultant *Npr1* gene-knockout phenotypes in mice reflect the critical roles of NPRA in physiological and pathophysiological processes occurring in hypertension and cardiovascular disorders. 

### 4.1. Effect of Npr1 Gene Ablation on Hypertension

The ANP/NPRA system plays important roles in the pathophysiology of hypertension and cardiovascular regulation. However, the precise mechanism of action of ANP/NPRA is still not well understood. Genetic studies of *Npr1* gene-deficient mice have provided evidence that NPRA deficiency increases BP in mutant mice as compared with wild-type animals [18,64,65,68,101,128]. Moreover, ablation of *Npr1* has indicated that the BP of homozygous mutant mice (0-copy) remains elevated and largely unchanged in response to either minimal- or high-salt diet, suggesting that mutations in *Npr1* may also explain salt-resistant form of hypertension [19,71,101,128]. Increase in circulating ANP/BNP is an important marker of cardiac hypertrophy and heart failure in both animal models and humans. Thus, increases in ANP and BNP are considered one of the most robust responses to hypertrophic factors in the heart. 

To examine whether the heart releases other natriuretic substances in response to volume expansion, *Npr1* gene-disrupted mice were volume-expanded by the infusion of an iso-oncotic solution [131]. In these mice, the release of ANP from the heart was shown by marked elevation in plasma ANP levels. Wild-type control mice responded to the volume expansion with diuresis and natriuresis; however, gene-disrupted mutant mice showed diminished diuresis and natriuresis in response to saline infusions [38,131,132]. Although early studies raised the question of whether or not saline infusion would contribute to hemodilution, it was possible that an exaggerated response might have been produced in saline-infused mice. In our studies, the administration of pure blood to anesthetized *Npr1* homozygous null mutant (0-copy), wild-type (2-copy), and gene-duplicated (4-copy) mice to produce intravascular volume expansion was not accompanied by hemodilution [65]. There was a moderate increase in plasma protein and hematocrit in both wild-type controls and *Npr1* null mutant mice, indicating that no hemodilution occurred with pure blood volume expansion in these animals. In wild-type control mice, urine flow and sodium excretion were associated with rises in glomerular filtration rate (GFR), but the MAP did not increase. In contrast, *Npr1* 0-copy mutant mice exhibited only a small change in urine flow and sodium excretion, in spite of the greater increases in MAP [65]. These results established that NPRA is essential for natriuretic and diuretic action in kidneys, mediating renal hemodynamic changes in response to acute pure blood volume expansion. 

The ablation of *Npr1* increases BP by 35–45 mmHg in *Npr1^−/−^* (0-copy) mutant mice as compared with *Npr1^+/+^* (2-copy) wild-type mice [18,64,66,68]. However, increased copy number of *Npr1* in gene-duplicated *Npr1^++/+^ and Npr1^++/++^* (3-copy and 4-copy) mice significantly reduced BP, corresponding to the increasing number of *Npr1* gene copies [19,65,71,73,77]. Our early studies examined the mechanisms mediating the responsiveness of varying numbers of *Npr1* gene copies by determining urine flow, sodium excretion, renal plasma flow (RPF), GFR, and BP in *Npr1* 0-copy, 2-copy, and 4-copy mice in a *Npr1* gene-dose-dependent manner [19,65,69,73,74,77]. Volume expansion with whole blood infusion significantly increased MAP in *Npr1* gene-disrupted (0-copy) mice compared with wild-type (2-copy) and gene-duplicated (4-copy) mice [65]. In 0-copy mice, the GFR was lower by almost 35% and higher by almost 45% in 4-copy mice than in 2-copy control mice. Similarly, RPF was lower by 30% in 0-copy and higher by almost 60% in gene-duplicated 4-copy mice as compared with 2-copy control animals. The 0-copy mice retained significantly higher levels of sodium and water than did other mice; however, gene-duplicated 4-copy mice showed drastically lower levels of both sodium and water than did 2-copy mice. Significant increases in plasma creatinine concentrations and urinary albumin excretion, together with reduced creatinine clearance rates, suggested the onset of renal insufficiency in *Npr1* 0-copy mutant mice. These results demonstrated that the ANP/NPRA/cGMP signaling axis is predominantly responsible for mediating the renal hemodynamic and sodium excretory in responses to intravascular blood volume expansion. Our early studies showed that ablation of *Npr1* in mice provokes kidney fibrosis, remodeling, and significant expression of pro-inflammatory and pro-fibrotic cytokines, as well as several genes participating in the nuclear factor kappa B (NF-κB) pathway [68,69]. The treatment of *Npr1* 0-copy mice with NF-κB inhibitors significantly reversed the renal hypertrophic and fibrotic responses in *Npr1* 0-copy animals [69].

### 4.2. Effect of Npr1 Gene Disruption on the Pathophysiology of Cardiac Dysfunction

Gene-disrupted *Npr1* mutant mice lacking NPRA showed marked cardiac hypertrophy and chamber dilation disproportionate to their increased BP [64,66,73]. Echocardiographic analysis of mice showed a compensated state of systemic hypertension in which cardiac hypertrophy and chamber dilation were evident, but there was no reduction in ventricular performance. Overall, the heart weight (HW) and echocardiographic data of *Npr1* gene-knockout mutant mice were accompanied by marked cardiac hypertrophy and ventricular enlargement [66,67,73,77]. Extensive cardiac hypertrophy in human seems to be associated with local ischemia that leads to the death of myocytes and cardiac fibrosis. Histological examination and morphometric comparisons has shown concentric hypertrophy with an increase in myocyte cross-sectional area in 0-copy null mutant male mice as compared with wild-type 2-copy animals [64,66,73]. Perivascular fibrosis has often been seen in male null mutant 0-copy mice; however, the extent of this pathology was somewhat variable [64,67]. The degree of cardiac hypertrophy and fibrosis was lower in female 0-copy mutant mice than in males [64,101]. Cardiac output and basal stroke calculations from echocardiography and hemodynamic data analysis suggested that the hearts of *Npr1* 0-copy mutant mice seem to work at least double as hard as those of 2-copy mice. 

The signaling mechanisms that mediate the development of cardiac hypertrophy in vivo still are not fully understood. To examine whether the ANP-BNP/NPRA pathway directly modulates hypertrophic responses in the heart, both *Npr1* gene-knockout and wild-type mice were subjected to pressure overload (60 mmHg) induced by transverse aortic constriction (TAC) method [106]. In 0-copy mutant mice, TAC induced left ventricular weight/body weight (LV/BW) ratio, left ventricular end diastolic dimension (LVEDD) and significant reduction in cardiac function as compared with 2-copy mice [106]. These authors also observed that chronic treatment of *Npr1* 0-copy mice with enalapril, which is an angiotensin-converting enzyme (ACE) inhibitor, furosemide, which is a diuretic, or losartan, an angiotensin II type 1 receptor (AT1R) blocker, effectively reduced BP to normal levels, although their HW/BW ratios were not decreased. We observed that ACE and AT1R are upregulated in *Npr1* 0-copy mice as compared with 2-copy wild-type mice [67]. These findings suggested that the ANP-BNP/NPRA signaling mechanism exerts direct antihypertrophic effects in the heart, independent of the BP control. The deletion mutations in *Npr1* gene have been suggested to reduce the receptor activity of NPRA and were considered as a potential genetic factor for hypertension and cardiac hypertrophy in humans [133]. 

### 4.3. Effect of Npr1 Ablation on the Renin-Angiotensin System and Cardiovascular Disorders

There is evidence that the chronic hypotensive effect of ANP/NPRA is partly mediated by suppressing RAAS and that ANP/NPRA opposes ANG II-mediated vascular and renal effects [4] Pandey 2008; Pandey 2018. In the absence of the counter-regulatory effect of ANP-BNP/NPRA signaling, sensitization of BP might result, at least in part, from failure to overcome the effects of RAAS. Early findings led to the notion that ANP plays an important role in regulating renal function by its direct vasodilatory effect and natriuretic response, as well as the antagonistic action of RAAS [6,9,17,19]. Our previous studies showed that renin and Ang II levels were significantly elevated in newborn male *Npr1* gene-knockout mice, while both renal and circulating renin levels were drastically reduced in adult male knockout mice [18]. However, the adrenal renin contents in adult male mice remained significantly elevated, suggesting that inhibition of renin and Ang II levels is a compensatory response to increased BP in adult homozygous null mutant mice. Thus, NPRA may differentially regulate adrenal versus renal renin and Ang II levels, resulting in increased absorption of salt and water through the kidneys, which is critical for the development of hypertension and cardiac dysfunction in *Npr1* null mutant mice [18,19]. 

Ablation of the ANP/NPRA signaling system inhibits aldosterone synthesis and its release from adrenal glomerulosa cells [18,19]. This may account for this system’s ANP-dependent renal natriuretic and diuretic effects. Our studies of *Npr1* gene-knockout mice suggested that at birth, the absence of NPRA allowed greater renin and Ang II levels and increased mRNA expression of renin [18,19]. However, when 0-copy mice reached 4–12 weeks of age, their circulating renin and Ang II levels were greatly reduced as compared to levels in 2-copy wild-type mice. We predicted that the decrease in renin content in adult *Npr1* 0-copy mice is most likely a result of progressive elevation of BP leading to inhibition of the synthesis of renin and its release from renal juxtaglomerular cells [65]. On the other hand, renin mRNA levels and renin contents, as well as both Ang II and aldosterone levels, were elevated in adult 2-copy mice compared with 0-copy mice [18,19]. 

Evidence suggests that the ANP/NPRA/cGMP system opposes the physiological and pathological actions of Ang II [6,134]. Our studies have shown that adrenal Ang II and aldosterone levels are decreased in *Npr1* gene-duplicated mice fed a high-salt diet, whereas a low-salt diet stimulated Ang II and aldosterone levels in the adrenal glands of both 0-copy and 4-copy mice [19,101]. On the other hand, a high-salt diet suppressed adrenal Ang II and aldosterone levels in 0-copy and 2-copy mice, but not in *Npr1* gene-duplicated 3-copy and 4-copy mice. These findings indicated that ANP/NPRA signaling is protective against high salt levels in *Npr1* gene-duplicated mice as compared with *Npr1* gene-knockout mice [19,101]. 

## 5. Consequences of Genetic Duplication of *Npr1*

Gene-duplication strategy by homologous recombination produces offspring, including 2-copies (1/1; wild-type), 3-copies (2/1), or 4-copies (2/2) of the target gene [61,71,72]. Method for duplicating genes by double-strand gap repair has also been reported [63]. In double-strand gap repair, the target gene acts as a template to fill in a gap between two regions of homology of the gene with a single crossover event to produce duplication of the target gene. Mice with 2, 3, or 4 copies of *Npr1* were produced to establish a direct gene-dose-dependent effect of *Npr1* on BP, cardiac hypertrophy, and inflammatory responses [19,65,73,77]. To determine whether mice with different *Npr1* copies are able to maintain stable physiological functions, *Npr1* gene-duplicated mice were used to examine the effect of changes in expression of the *Npr1* gene copy numbers on BP and cardiovascular disorders [19,71,73,101]. 

Our results also showed that changes in the *Npr1* copy number, with corresponding changes in expression of NPRA, causes progressive changes in BP in mice kept for three weeks on a low-salt diet (0.05% NaCl), intermediate-salt diet (2.0% NaCl), or high-salt diet (8% NaCl) [19,71,101]. Salt had a significant effect on BP in 1-copy mice, but only a modest effect in mice with 2-copies of *Npr1*. However, BP in animals having 3- and 4-copies of *Npr1* were negatively affected by increasing dietary salt levels. These results demonstrated that below-normal *Npr1* expression leads to salt-sensitive increases in BP, whereas above-normal *Npr1* expression lowers BP and protects against high dietary salt concentrations [19,71]. 

## 6. Genetic Disruption of *Npr1* and Cardiac Hypertrophy, Fibrosis, and Inflammation

Genetic ablation of *Npr1* in mice increases cardiac mass and the incidence of hypertrophy, fibrosis, and inflammation [64,66,67,73,77,100,101,135,136,137]. *Npr1* gene-knockout mice develop cardiac hypertrophy, fibrosis, and inflammatory responses independent of BP [66,67,138]. We have demonstrated that *Npr1* gene disruption in mice not only increases the expression of cardiac hypertrophic markers, pro-inflammatory cytokines, matrix metalloproteinases (MMP-2, MMP-9), and markers of fibrosis, but also increases the expression and activation of two transcription factors, NF-κB, and activating protein-1 (AP-1) [66,67,73,77,101]. Both of these transcription factors have been found to be associated with cardiac hypertrophy, fibrosis, and extracellular matrix remodeling in *Npr1* gene-disrupted mouse models [7,10,66,67,73]. Moreover, *Npr1* gene-disruption activates NF-κB and AP-1, leading to cardiovascular disease conditions. 

ANP/NPRA signaling antagonizes Ang II-induced collagen systhesis by suppressing the activities of MMP-2, MMP-9, and nuclear translocation of NF-κB [73,139]. The expression of angiotensin-converting enzyme (ACE) and AT1R is significantly increased in *Npr1* null mutant mice compared with wild-type mice [67,138]. However, the expression of sarcolemal/endoplasmic reticulum Ca^2+^-ATPase-2a (SERCA-2a) was progressively decreased in the hypertrophied hearts of *Npr1* gene-ablated mice [66,138]. 

ANP/NPRA signaling antagonizes Ang II and AT1R-mediated cardiac remodeling and defects; it also provides a protective mechanism in hypertrophied and failing hearts [67,73,77,101,140,141]. Selective inactivation of *Npr1* by homologous lox/Cre-mediated recombination led to only mild cardiac hypertrophy, but ANP levels were significantly increased [55,142,143,144,145]. In *Npr1* gene-knockout mice, both ANP and BNP levels were markedly increased. These mice showed a higher incidence of congestive heart failure and significantly greater mortality than did wild-type mice [64,66,67,135]. 

## 7. Association of *Nppa, Nppb,* and *Npr1* Polymorphisms with Cardiovascular Dysfunction

Genetic and clinical studies have demonstrated the association of single nucleotide polymorphisms (SNPs) of *Nppa*, *Nppb*, and *Npr1* polymorphisms with cardiovascular dysfunction in humans [107,133,146,147,148,149,150,151]. It is known that patients with monogenic forms of hypertension have rare genetic mutations [152,153]. Various genes among the pathways, including natriuretic peptides, RAAS, and the adrenergic systems, have been found to regulate inherited multigenic disease traits like hypertension and cardiovascular disorders. Nevertheless, the genetic determinants in these pathways that contribute to inter-individual differences in BP regulation have been linked only with the NPs and their receptor systems [146]. Moreover, SNPs of *Nppa*, *Nppb*, and *Npr1* have been found to be associated with increased circulating ANP and BNP, lower systolic and diastolic BP, and decreased prevalence of cardiometabolic syndrome [146,154,155]. Interestingly, an association between *Nppa* promoter polymorphism and cardiac hypertrophy has been demonstrated in hypertensive Italian patients, suggesting that patients carrying SNPs of *Nppa* have marked decreases in pro-ANP levels and left ventricular hypertrophy [147]. 

The genetic association between the microsatellite marker in *Npr1* promoter and left ventricular hypertrophy indicated that the ANP-BNP/NPRA signaling system contributes to cardiac remodeling and congestive heart failure in patients with essential hypertension [147]. The relationship between high BP and cardiovascular risk exists; thus, in the absence of ANP-BNP/NPRA/cGMP signaling, a small increase in BP may cause the detrimental consequences of cardiac events, including congestive heart failure. 

It has been suggested that the underlying mechanism of high BP and cardiovascular events could be linked with *Npr1* mRNA instability, and that this leads to a decrease in translational products of receptor protein [156]. Consequently, a feedback mechanism could be elicited whereby diminished function of ANP-BNP/NPRA/cGMP signaling caused by the defect in the *Npr1* gene could, in compensation, trigger increased expression and release of ANP and/or BNP into the circulation. Earlier studies also indicated that substantial heritability of BP and cardiovascular risks might play a role for genetic factors [157]. It has been shown that SNP in human *Npr1* significantly decreases the expression of *Npr1,* suggesting that a single allele mutation may be associated with increased susceptibility to hypertension and cardiac hypertrophy [133]. In humans, a “4-minus” haplotype in the 3′-non-coding region of the *Npr1* promoter was found to be linked with elevated NT-proBNP levels [133,148]. Thus, it is thought that in humans there is a positive association between *Nppa*, *Nppb*, and *Npr1* gene polymorphisms and high BP, cardiac hypertrophy, and heart failure. The variations in SNPs of *Nppa*, *Nppb*, and *Npr1* in human patients seem to be associated with a family history of high BP and cardiovascular disorders, including myocardial infarction, cardiac mass, septal wall thickness, and congestive heart failure. Because common variants associated with BP and heart failure have not yet been fully identified, further studies are needed to characterize the most functionally significant markers of *Npp*a, *Nppb,* and *Npr1* variants in a large patient population. Interestingly, it has been shown that the minor allele of the ANP genetic variant rs5068 is associated with high circulating levels of ANP and BNP, low risk of hypertension, high plasma levels of high-density lipoprotein (HDL) cholesterol, and low prevalence of metabolic syndrome and obesity [155].

## 8. Therapeutic Use of Natriuretic Peptides

The physiological and pathophysiological functions of ANP, BNP, and NPRA signaling are associated with protective mechanisms in various organ systems, including the heart, kidneys, central nervous system, lung, and vasculature. The injection of recombinant ANP (Carperitide) was found to accelerate the recovery of blood flow with increased capillary density in ischemic and hyperglycemic conditions [158,159]. Recombinant BNP (nesiritide) also elicited natriuretic and vasodilatory action in heart failure patients [160]. Both ANP and BNP have been shown to exert antihypertrophic and antifibrotic functions [161]. Laser Doppler perfusion studies of *Npr1* gene-knockout mice have shown that blood flow recovery in ischemic limbs is markedly inhibited [162]. Critical hind limb ischemia also severely impaired these animals [163]. 

The key elements regulating BP and cardiovascular homeostasis in humans and mice are very similar in all of their main attributes related end-organ damage. Given the apparent action of ANP-BNP/NPRA system in antagonizing the RAAS, there is great potential in harnessing the ANP-BNP/NPRA axis as a novel therapeutic intervention in hypertension and cardiovascular diseases. Thus far, clinical trials have identified both benefits and risks of synthetic ANP (anaritide, carperitide) and BNP (nesiritide) for treating hypertension, renal insufficiency, and heart failure in humans; however, these drugs have not yet joined the mainstream of therapeutic treatments in the US [164,165,166]. Alternatively, enhancing endogenous levels of ANP-BNP/NPRA signaling by inhibiting neprilysin has recently been proven to be a critical component in the first new therapeutic targets for hypertension and heart failure to have been developed in decades [167]. 

## 9. Current and Future Perspectives

In about the last four decades, a large body of information has provided a unique perspective on delineating cellular, biochemical, molecular, and genetic information on the regulation and function of natriuretic peptides and their receptor system in the regulation of both the physiology and pathophysiology of cardiovascular homeostasis. Gene-targeting methods (knockout and gene duplication) have been used to delineate the genetic functions dictated by decreasing and/or increasing numbers of *Npr1* gene copies in mice in a gene-dose-dependent manner. Analyses of the molecular genetics and physiological and pathological phenotypes of *Npr1* gene-targeted mutant mice have defined the pivotal roles of ANP/NPRA/cGMP signaling in disease states by genetically modified *Npr1* gene copy numbers and their protein product levels in intact animals. An intriguing finding was that a common genetic variant at the *Nppa* and *Nppb* locus is associated with circulating concentrations of ANP and BNP concentrations, contributing to inter-individual variations in cardiovascular homeostasis. Future investigations should provide a better understanding of the genetic basis of *Npr1* in regulating hypertension, strokes, and cardiovascular disorders. 

ANP and BNP are considered to be critical markers of congestive heart failure, but their therapeutic potential in the treatment of cardiovascular diseases such as high BP, renal failure, congestive heart failure, and stroke remains to be established. Ongoing molecular and genetic investigations of *Nppa*, *Nppb*, and *Npr1* should be of great value in resolving the genetic complexities related to hypertension and cardiovascular events, including congestive heart failure. Overall, current and ongoing and future studies should be directed toward providing a unique perspective on delineating the genetic, molecular, and physiological basis of *Nppa*, *Nppb*, and *Npr1* gene expression and regulation under physiological and pathological conditions in both normal and disease states in both experimental animals and humans. 

The results of future investigations should provide new therapeutic targets for preventing and treating cardiovascular diseases, stroke, and other abnormal cardiac conditions. Above all, clinical studies are needed to characterize more functionally significant markers of *Nppa, Nppb*, and *Npr1* variants in large human populations. In the clinical environment, human recombinant ANP and BNP could possibly be used for heart failure therapy, but first more careful molecular and genetic investigations are needed. Future progress in this field will significantly strengthen and advance our knowledge of genetic and molecular approaches to evaluate the diverse physiological and pathophysiological functions related to cardiovascular disorders. In the future, we should expect to define the potential clinical implications of ongoing investigations of the pharmacogenomics of *Nppa*, *Nppb*, and *Npr1*. Right now we are just at the initial stage of the exponential phase of molecular therapeutics and genomic advancement of the functional aspects of natriuretic peptides and their receptor systems for use in the treatment, prevention, and control of hypertension, heart failure, and other cardiovascular and neurological disorders. More investigations are needed to extend this possibility.

## Figures and Tables

**Figure 1 ijms-20-03946-f001:**
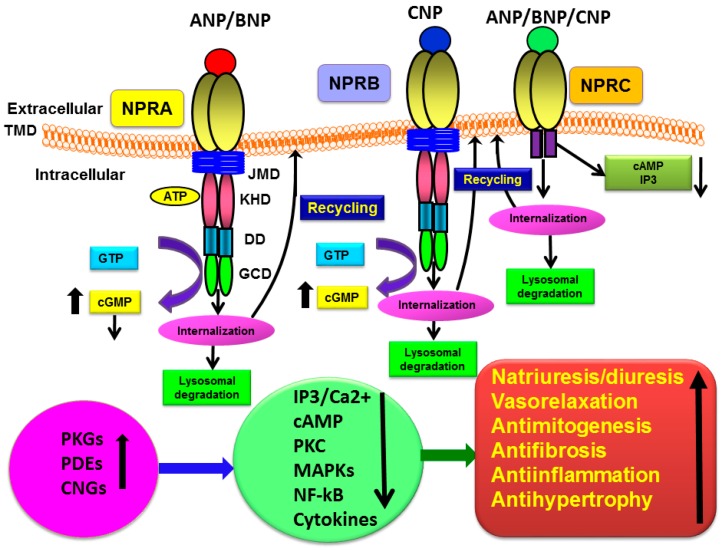
Diagrammatic representation of ligand-stimulation of ligand–receptor complex activation and the physiological functions of natriuretic peptide receptor-A (NPRA), receptor-B (NPRB), and receptor-C (NPRC): Ligand-binding activates NPRA and NPRB, which leads to enhanced production of the intracellular second messenger cGMP, as well as with stimulation and activation of ligand-dependent cellular and physiological responsiveness. CNP activates NPRB and all three NPs activate NPRC. The bound-ligand receptor complexes of NPRA, NPRB, and NPRC are rapidly internalized and a large population of receptors is degraded in lysosomes. However, a small population of ligand-bound receptors is recycled back to the plasma membranes. ANP, atrial natriuretic peptide; BNP, brain natriuretic peptide; CNP, C-type natriuretic peptide; LBD, ligand binding domain; TMD, transmembrane domain; JMD, juxtamembrane domain; KHD, kinase like homology domain; DD, dimerization domain; GCD, guanylyl cyclase catalytic domain; IP_3_, inositol trisphosphate; PKG, cGMP-dependent protein kinase or protein kinase G; PDE, phosphodiesterase; CNG, gated-ion channel; MAPKs, mitogen-activated protein kinase; NF-κB, nuclear factor kappa B.

**Table 1 ijms-20-03946-t001:** The gene-knockout disease-specific phenotypes of mice with gene-disruption of natriuretic peptides and their cognate receptors.

Gene	Peptide/Protein	Gene-Knockout
Nomenclature	Nomenclature	Phenotype in Mouse
*Nppa*	ANP	Volume overload, high blood pressure and hypertension, salt sensitivity, fibrosis, and cardiac disorders [80,82,83,84]
*Nppb*	BNP	Hypertension, sodium excretion, vascular complication, and fibrosis [81,124]
*Nppc*	CNP	Dwarfism, reduced bone growth, and impaired endochondral ossification [125,126,127]
*Npr1*	NPRA	High blood pressure and hypertension, salt sensitivity, volume overload, and cardiac hypertrophy, fibrosis, and inflammation [64,65,66,73,77,97,100,128]
*Npr2*	NPRB	Dwarfism, seizures, female sterility, and decreased adiposity [129,130]
*Npr3*	NPRC	Bone deformation and long bone overgrowth [62]

Nomenclature of gene and peptide or protein with gene-knockout phenotype. The gene nomenclature is indicated in italic and peptide or protein is indicated in capital letters. *Nppa,* coding for pro-ANP; *Nppb*, coding for pro-BNP; *Nppc,* coding for pro-CNP; ANP, atrial natriuretic peptide; BNP, brain natriuretic peptide; CNP, C-Type natriuretic peptide; *Npr1* (coding for NPRA); *Npr2*, (coding for NPRB); *Npr3* (coding for NPRC). NPRA, natriuretic peptide receptor-A; NPRB, natriuretic peptide receptor-B; and NPRC, natriuretic peptide receptor-C.

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
