# Peer review of "Genetic Ablation and Guanylyl Cyclase/Natriuretic Peptide Receptor-A: Impact on the Pathophysiology of Cardiovascular Dysfunction"

_ijms, 2019, doi:10.3390/ijms20163946_

Round 1

Reviewer 1 Report

The review by K Pandey focuses on the natriuretic peptide receptor NPR1 and the effects of NPR1 gene duplication or knockout on cardiovascular pathophysiology. This review is interesting and is generally well written.

Specific Comments.

Line 67 briefly mentions NPRC. More information should be included here – for instance, that NPRC functions as a clearance receptor, internalizing and degrading bound natriuretic peptide. Also, make mention the possibility of signaling via an inhibitory G protein regulating adenyl cyclase activity (Fig 1).

Line 114.  Is D-type natriuretic peptide a unerversially agreed abbreviation? It is more usually referred to as Dendroaspis natriuretic peptide (DNP) following its isolation from venom of the green Mamba snake Dendroaspis angusticeps. There is scant evidence that DNP is present in humans – DNP assays have not been vigorously validated.

Line 119. Mentions of NEP. Further discussion on the physiological clearance mechanisms of natriuretic peptides from the circulation and extracellular fluids which should be included in the introduction. These should include discussion relating to neprilysin (NEP), insulin degrading enzyme (IDE), the natriuretic peptide clearance receptor and osteocrin a competitive ligand for NPRC.

Lines 307-310. The two sentences seem to say the same thing – with ablation of Npr1, blood pressure is elevated (chronically), including in the state of a high salt diet. Yet the second sentence starts with “In contrast … and then almost paraphrases the first sentence – “… ablation of Npr1 results in chronic elevation of BP …”

Author Response

Reviewer 1:

I greatly appreciated the reviewer remarks that this review is interesting and is generally well written.

Comment 1: Line 67 briefly mentions NPRC. More information should be included here – for instance, that NPRC functions as a clearance receptor, internalizing and degrading bound natriuretic peptide. Also, make mention the possibility of signaling via an inhibitory G protein regulating adenyl cyclase activity (Fig 1).

Response: In the present review, the emphasis has been placed on the physiological and pathophysiological significance of gene-targeting (gene-knockout and gene-duplication) of Npr1 (coding for NPRA) to delineate the impact and mechanisms regulating the cardiovascular homeostasis. Adding the discussion on NPRC would make this review lengthy and more descriptive. In accordance with the reviewer’s comments, the internalization and recycling arrows for NPRA, NPRB, and NPRC have been added in Figure 1 to indicate that NPRs are internalized and a majority of the ligand-bound receptors are degraded in the lysosomal compartments in the revised manuscript (lines 67-71).

Comment 2: Line 114.  Is D-type natriuretic peptide a unerversially agreed abbreviation? It is more usually referred to as Dendroaspis natriuretic peptide (DNP) following its isolation from venom of the green Mamba snake Dendroaspis angusticeps. There is scant evidence that DNP is present in humans – DNP assays have not been vigorously validated.

Response: As suggested by the reviewer, the phrase “D-type natriuretic peptide” has been removed and “Dendroaspis natriuretic peptide (DNP)” has been added in the revised manuscript (lines 120, 121).

Comment 3: Line 119. Mentions of NEP. Further discussion on the physiological clearance mechanisms of natriuretic peptides from the circulation and extracellular fluids which should be included in the introduction. These should include discussion relating to neprilysin (NEP), insulin degrading enzyme (IDE), the natriuretic peptide clearance receptor and osteocrin a competitive ligand for NPRC.

Response: As suggested by the reviewer, the clarification has been made regarding the significance of inhibiting the neutral endopeptidase (NEP) in enhancing the endogenous levels of ANP-BNP/NPRA signaling. In this context, the previous statement in lines 527-529 of the original manuscript regarding the role of inhibiting neprilysin has been more clearly stated in the revised manuscript (lines 594, 595). 

Comment 4: Lines 307-310. The two sentences seem to say the same thing – with ablation of Npr1, blood pressure is elevated (chronically), including in the state of a high salt diet. Yet the second sentence starts with “In contrast … and then almost paraphrases the first sentence – “… ablation of Npr1 results in chronic elevation of BP …”

Response: As suggested by the reviewer, the redundant sentence in lines 307-310 has been deleted and the modified sentence has been more clearly stated. The redundant sentence indicating “in contrast, the ablation of Npr1 resulted in chronic elevation of BP in mice fed high-salt diet” has been deleted in the revised manuscript (lines 344, 345).

Reviewer 2 Report

The present review article put an overwhelming number of references together focusing on the impact of natriuretic receptor-A (Npr1) gene ablation on the pathogenesis such as hypertension, kidney disorders, and cardiovascular dysfunction.  By referring to mouse models carrying gene disruption and gene duplication, Pandey KN described Npr1 gene mediates the role of NPs and NPRs in dose-dependently.  This review can’t bring an entirely new concept, but it may provide a genetic consensus of Npr1 physiology in hypertension and heart diseases.  The following matters would be additionally considered for the comprehensive insights especially to the role of GC-A/Npr1.

The author mentioned the effect of tissue-specific gene ablation in the abstract, but there is no explanation with references which show the phenotypes of Npr1 using developed materials such as the conditional gene knockout in a certain tissue and at a specific time. Give an explanation and cite the updates related to the specific impact of Npr1 with target specific gene knockout for vascular endothelial cells, vascular smooth muscle cells, or cardiac myocytes.  GC-A/NPRA activation generates intracellular second-messenger cyclic GMP to regulate a broad array of physiological processes in cardiovascular system. As shown in the introduction, both NPRA and NPRB can produce cGMP, while nitric oxide also increases cGMP synthesis via soluble GC activation in various cells.  However, the concentration of cGMP levels and the intracellular compartmentation are known to be well organized partly with localized phosphodiesterases.  The actions of cGMP and the downstream signaling pathways are also known to result in mediating diverse functions in widespread tissue even when the activation of same receptor occurs.  It would be helpful to refer to the discrepancy of physiological and pathophysiological functions of Npr1/cGMP in cardiovascular tissue between Nppa and Nppb, by adding insights from the viewpoint of differential downstream signaling pathways.  Figure 1 gives an overview of three types of NPRs signaling. GC-A/NPRA dependent signaling pathway is displayed in color-coding of the diagrammatic representation.  However, the classified signaling pathways were not given an explanation in the text, while non-target of NPs and NPRs was kindly clarified (e.g. Nppc, CD-NP).  As a review article of NPRA, it should provide an explanation for Npr1 dependent signaling in detail, which can also help understand the main figure.  The number of references can be reduced by carefully selecting recent significant original articles to each finding.  It would be great if you could discriminate the significance of this article to previous review ones.

Author Response

Reviewer 2:

I greatly appreciated the reviewer’s comments that the present review has focused on the impact of Npr1 gene ablation on the pathogenesis of hypertension, kidney diseases, and cardiovascular dysfunction.

Comment 1: The author mentioned the effect of tissue-specific gene ablation in the abstract, but there is no explanation with references which show the phenotypes of Npr1 using developed materials such as the conditional gene knockout in a certain tissue and at a specific time.

Response: The current review has specifically focused on the significance of global gene-knockout and gene-duplication of Npr1 in regulating the cardiovascular homeostasis in a gene-dose-dependent manner. To keep the manuscript in succinct format, the Npr1 ablation in conditional knockout, which seem to be controversial to some extent, has been avoided in the current review. More studies are needed to discuss these issues in future review manuscript.

Comment 2: As shown in the introduction, both NPRA and NPRB can produce cGMP, while nitric oxide also increases cGMP synthesis via soluble GC activation in various cells.

Response: There is no report regarding the compensational effects of NPRA and NPRB system with nitric oxide system on any of the physiological effects. Therefore, the issue of nitric oxide and soluble guanylyl cyclase was not discussed in the current manuscript.

Comment 3: The actions of cGMP and the downstream signaling pathways are also known to result in mediating diverse functions in widespread tissue even when the activation of same receptor occurs.

Response: The goal of the present manuscript was not to explore general biology of NPRA and NPRB. The current review has focused on the significance of the receptor NPRA and its gene Npr1 in the pathobiology of cardiovascular disease states. The physiological functions of NPRA has been more clearly depicted in Figure 1 of the revised manuscript (lines 68-83).

Comment 4: Figure 1 gives an overview of three types of NPRs signaling. GC-A/NPRA dependent signaling pathway is displayed in color-coding of the diagrammatic representation.  

Response: Figure 1 depicts only the major physiological functions of NPRA in succinct manner in relation to the cardiovascular functions. The current review was intended to provide a succinct report on the physiological and pathophysiological significance of ANP/NPRA/cGMP signaling pathway in cardiovascular homeostasis. The Figure 1 has been revised and developed to indicate the internalization, recycling, and degradation of all three receptors, including NPRA, NPRB, and NPRC (lines 68-83).

Comment 5: The number of references can be reduced by carefully selecting recent significant original articles to each finding. 

Response: A great deal of studies have been conducted, describing the role of ANP/NPRA/cGMP system regulating the physiology and pathophysiology of the cardiovascular system in relation to cardiac defects and heart failure. In the current review, the effort was made to limit the references. In the interest of space and Journal style, still a large number of important studies could not be afforded in the current manuscript.

Comment 6: It would be great if you could discriminate the significance of this article to previous review ones.

Response: The current manuscript has mainly focused on the role of NPRA and its gene Npr1, regulating the cardiovascular effects; however, previous articles have largely discussed in general the effects of NPRA, NPRB, and NPRC in the renal, vascular, cardiac, and endocrine systems.

Round 2

Reviewer 2 Report

The author handled the reviewer’s concerns, but the point is that the current review doesn’t bring any new to the field of NPR-A/Npr1 especially in the pathobiology of cardiovascular diseases.  As study limitations, all discussion has been conducted only in global gene knockout of Npr1 and the gene duplication, where it doesn’t always help understand the direct effect of Npr1 gene particularly in cardiovascular system on which the author was going to focus.  it’s acceptable if any new specific concept can be raised to contribute to our comprehensive understanding of the gene.